# Cytotoxicity of *Vibrio parahaemolyticus* AHPND toxin on shrimp hemocytes, a newly identified target tissue, involves binding of toxin to aminopeptidase N1 receptor

Waruntorn Luangtrakul[1], Pakpoom Boonchuen[1,2], Phattarunda Jaree[3], Ramya Kumar[4,5], Han-Ching Wang[4,5]*, Kunlaya Somboonwiwat[1]*

**1** Center of Excellence for Molecular Biology and Genomics of Shrimp, Department of Biochemistry, Faculty of Science, Chulalongkorn University, Bangkok, Thailand, **2** School of Biotechnology, Institute of Agricultural Technology, Suranaree University of Technology, Nakhon Ratchasima, Thailand, **3** Center of Applied Shrimp Research and Innovation, Institute of Molecular Biosciences, Mahidol University, Nakhon Pathom, Thailand, **4** Department of Biotechnology and Bioindustry Sciences, College of Biosciences and Biotechnology, National Cheng Kung University, Tainan, Taiwan, **5** International Center for the Scientific Development of Shrimp Aquaculture, National Cheng Kung University, Tainan, Taiwan

* wanghc@mail.ncku.edu.tw (HC); kunlaya.s@chula.ac.th (KS)

## Abstract

Acute hepatopancreatic necrosis disease (AHPND) caused by PirAB$^{VP}$-producing strain of *Vibrio parahaemolyticus*, VP$_{AHPND}$, has seriously impacted the shrimp production. Although the VP$_{AHPND}$ toxin is known as the VP$_{AHPND}$ virulence factor, a receptor that mediates its action has not been identified. An in-house transcriptome of *Litopenaeus vannamei* hemocytes allows us to identify two proteins from the aminopeptidase N family, *Lv*APN1 and *Lv*APN2, the proteins of which in insect are known to be receptors for Cry toxin. The membrane-bound APN, *Lv*APN1, was characterized to determine if it was a VP$_{AHPND}$ toxin receptor. The increased expression of *Lv*APN1 was found in hemocytes, stomach, and hepatopancreas after the shrimp were challenged with either VP$_{AHPND}$ or the partially purified VP$_{AHPND}$ toxin. *Lv*APN1 knockdown reduced the mortality, histopathological signs of AHPND in the hepatopancreas, and the number of virulent VP$_{AHPND}$ bacteria in the stomach after VP$_{AHPND}$ toxin challenge. In addition, *Lv*APN1 silencing prevented the toxin from causing severe damage to the hemocytes and sustained both the total hemocyte count (THC) and the percentage of living hemocytes. We found that the r*Lv*APN1 directly bound to both rPirA$^{VP}$ and rPirB$^{VP}$ toxins, supporting the notion that silencing of *Lv*APN1 prevented the VP$_{AHPND}$ toxin from passing through the cell membrane of hemocytes. We concluded that the *Lv*APN1 was involved in AHPND pathogenesis and acted as a VP$_{AHPND}$ toxin receptor mediating the toxin penetration into hemocytes. Besides, this was the first report on the toxic effect of VP$_{AHPND}$ toxin on hemocytes other than the known target tissues, hepatopancreas and stomach.

**Data Availability Statement:** All relevant data are within the manuscript and its Supporting Information files.

**Funding:** This research was funded by Chulalongkorn University under the Ratchadaphisek Somphot Endowment (CU_GR_62_79_23_30) to KS and the Thailand Research Fund (International Research Network Scholar (No. IRN61W0001)) to KS. Student fellowships to WL from the 100th Anniversary Chulalongkorn University Fund for Doctoral Scholarship, the 90th Anniversary of Chulalongkorn University Fund, and the Overseas Research Experience Scholarship for Graduate Students from the Graduate School, Chulalongkorn University, are greatly appreciated. Additional support from the Ministry of Science and Technology, Taiwan (MOST-108-2314-B-006-096-MY3) to HC is also acknowledged. The funders had no role in study design, data collection and analysis, decision to publish, or preparation of the manuscript.

**Competing interests:** The authors have declared that no competing interests exist.

## Author summary

A specific strain of *Vibrio parahaemolyticus* causing acute hepatopancreatic necrosis disease (AHPND) in shrimp or VP$_{AHPND}$ produces a binary toxin (PirAB$^{vp}$ toxin) that is previously known to induce cell death of stomach and hepatopancreas but the molecular mechanism has not been defined. Similar to Cry toxin receptor in insects, a novel aminopeptidase N1 protein from *L. vannamei* (*Lv*APN1) was identified as a putative receptor of VP$_{AHPND}$ toxin. Suppression of *Lv*APN1 reduced the number of AHPND virulence plasmids in stomach and occurrence of AHPND clinical sign, sustained the number of total hemocyte count, and elevated the number of viable hemocyte. We demonstrated that VP$_{AHPND}$ toxin challenge induces hemocyte cell damage and it interacts with *Lv*APN1 *in vitro*. Collectively, our finding suggested that not only stomach and hepatopancreas but also hemocyte are the VP$_{AHPND}$ target tissues where *Lv*APN1 serves as a VP$_{AHPND}$ toxin receptor. This study provides novel insight into the contributions of *Lv*APN1 receptor towards the AHPND pathogenesis in shrimp and may extend to the development of AHPND preventive measure in shrimp.

## Introduction

Acute hepatopancreatic necrosis disease (AHPND), initially referred to as early mortality syndrome (EMS), has caused severe mortalities in farmed penaeid shrimp throughout Southeast Asia including China in 2009 before it spread to Vietnam in early 2011 and Thailand in late 2011 [1–3]. AHPND can cause up to 100% mortality within 30 days after stocking, and has also resulted in production losses of more than US $1 billion per year in the Asian shrimp farming industry [4,5]. The causative agent of AHPND was found to be a specific strain of the Gram-negative halophilic marine bacterium *Vibrio parahaemolyticus* [6]. AHPND-causing bacteria initially colonize in the stomach of infected shrimp [6,7] to produce observable symptoms that include lethargy, an empty stomach and midgut, and pale to white atrophied hepatopancreas. Histological analysis of the hepatopancreas reveals sloughing of tubule epithelial cells in the early stage of AHPND and massive hemocytic infiltration in the late stage of infection [6,8,9].

All AHPND-causing *V. parahaemolyticus* strains carry a virulent pVA1 plasmid (VP$_{AHPND}$) which encodes the binary toxins PirA$^{vp}$ and PirB$^{vp}$. These toxins are homologous to the *Photorhabdus luminescens* insect-related (Pir) toxins [10], and they are secreted into the extracellular environment. Reverse gavage experiments have shown that the bacteria-free supernatant of the bacterial culture is sufficient to induce typical AHPND symptoms [6], and in fact that AHPND-like symptoms can be produced by the reverse gavage injection of purified recombinant PirB$^{vp}$ toxin alone [10].

An analysis of the binary toxins crystal structure found that *V. parahaemolyticus* PirA$^{vp}$ and PirB$^{vp}$ form a heterodimer and that the overall structural topology of the PirAB$^{vp}$ toxins is very similar to that of *Bacillus thuringiensis* crystal insecticidal (Cry) toxin [10]. This similarity suggested that the putative PirAB$^{vp}$ heterodimer might have similar functional domains to the Cry protein, with the N-terminal domain of PirB$^{vp}$ corresponding to Cry domain I (pore-forming activity), the C-terminal domain of PirB$^{vp}$ corresponding to Cry domain II (receptor binding), and PirA$^{vp}$ corresponds to Cry domain III, which is thought to be related to receptor recognition and membrane insertion [11,12]. In the case of the *B. thuringiensis* Cry toxin, cell death is induced by a series of processes which include receptor binding, oligomerization and pore forming [12]. Briefly, the initial interaction is between domain III of Cry1A toxin and the

GalNAc sugar on the aminopeptidase N (APN) receptor, and this facilitates further binding of domain II to another region of the same receptor [10–12]. This binding promotes the localization and accumulation of additional molecules of the activated toxins. This assemblage of toxins then binds to another receptor, cadherin, and this promotes the proteolytic cleavage of the toxin's N-terminal α1 helix. This cleavage in turn induces the formation of the pre-pore oligomer and increases the oligomer binding affinity to the glycosylphosphatidylinositol (GPI)-anchored APN and alkaline phosphatase (ALP) receptors. The oligomer then inserts into the membrane, leading to pore formation and cell lysis [12]. The PirAB$^{vp}$ toxins are known to mainly target the shrimp hepatopancreas. Recently, it was found that, in the brine shrimp larvae, PirAB$^{vp}$ toxin challenge induced damage to the digestive tract upon binding to epithelial cells and produces characteristic symptoms of AHPND. The extensive necrosis and damages epithelial cells in the midgut and hindgut regions, resulting in pyknosis, cell vacuolisation, and mitochondrial and rough endoplasmic reticulum (RER) damage [13]. However, while the PirAB$^{vp}$ binary toxins is thought to use a mechanism that is similar to that of the Cry pore forming toxin, the details of this mechanism remain unclear. In the present study we therefore investigate the role of APN, which is one of the receptors that are already known to be central to this process in insects.

The APN family is composed of a class of zinc metalloproteinases that preferentially cleave single neutral amino acids from the N-terminus of polypeptides [14]. APNs are major proteins in the midgut of insect larvae, where they occur either as soluble enzymes or in association with the microvillar membrane [15]. APNs are involved in several functions in a wide range of species. For example, APNs in the lepidopteran larval midgut play an important role in the digestion of dietary protein [16]. The typical features present in classical lepidopteran APNs include a potential signal peptide at the N-terminus, a characteristic zinc-binding motif HEXXH(X)$_{18}$E essential for their enzymatic activity, a highly conserved GAMEN motif forming part of the active site and a GPI anchor signal sequence at the C-terminal attaching them to the membrane [14,17].

In the present study, we retrieved *Lv*APN1 and *Lv*APN2 sequences from our transcriptomic database of VP~AHPND~-challenged *Litopenaeus vannamei* hemocytes [18]. Gene expression of *Lv*APN1 was analyzed after challenge with AHPND-causing bacteria and partially purified VP~AHPND~ toxins. Next, the role and importance of the *Lv*APN1 receptor were investigated by using a gene silencing technique. Lastly, we provide a preliminary schematic representation that shows how *Lv*APN1 in hemocytes might be involved in AHPND pathogenesis.

## Results

### Characterization of the putative *Lv*APN protein

A comparison of the *Lv*APN1 and *Lv*APN2 deduced amino acid sequences showed that the encoded proteins both have high similarity to other aminopeptidases. While both *Lv*APNs have a Cry toxin binding region (CBR), the N-terminus of *Lv*APN1 but not *Lv*APN2 contains a transmembrane domain (Fig 1A). Considering the peptidase-M1 domain, both proteins have the HEXXH(X)$_{18}$E domain which is characteristic of the zinc-dependent metalloprotease gluzincins, as well as the GAMEN domain which is normally found in other APNs (Fig 1A). Also present is an ERAP1 domain, which plays a central role in N-terminal peptide trimming. The putative *N*-glycosylation sites and *O*-glycosylation sites were also identified (S1 Fig). No glycosylphosphatidylinositol (GPI) anchor sites were found in the C-terminal region, and no signal peptide was predicted.

Sequence alignment of the CBRs from *Lv*APNs and other APNs is shown in Fig 1B. The CBR located in the N-terminal region of the *Lv*APNs conformed to the specific CBR motif TxFxxTxARxAFPCxDEP that is found in the Cry-binding APNs of toxin-susceptible insects.

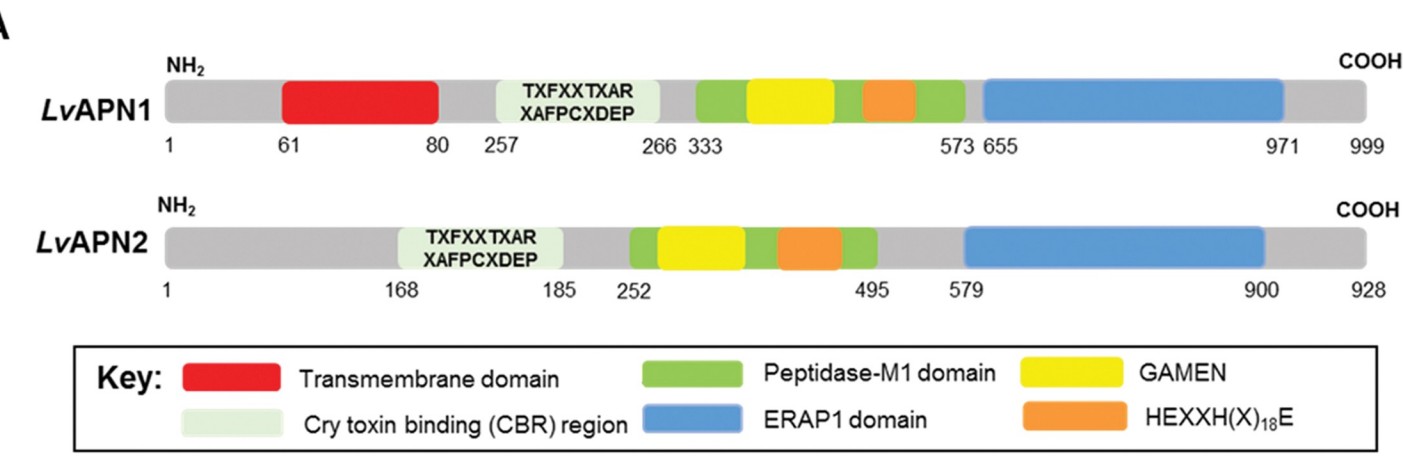

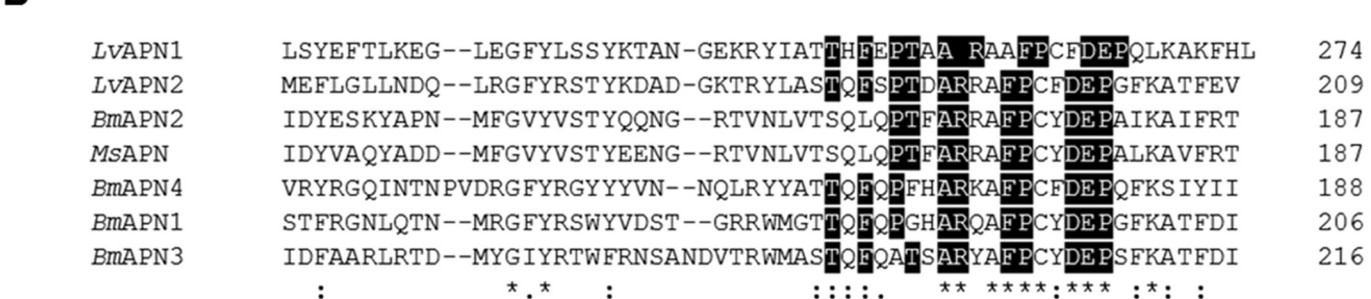

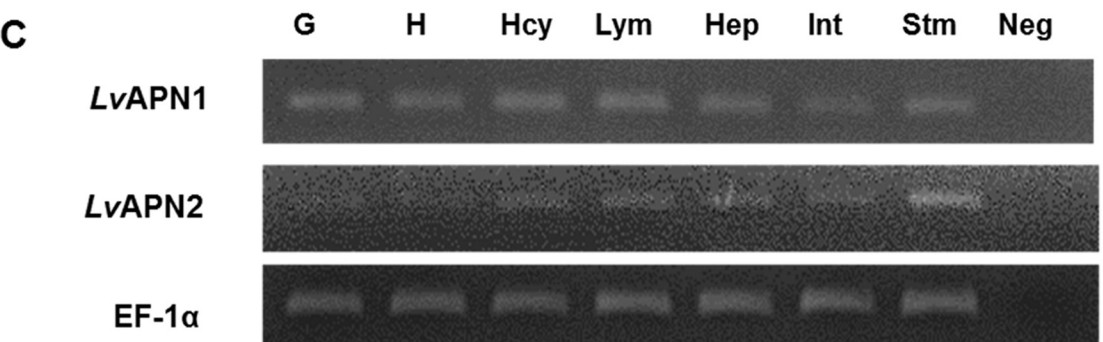

**Fig 1. *Lv*APN1 characteristics analysis.** (A) Schematic presentation of specific motifs and other components in the *Lv*APN sequence. Predicted positions of the putative N-terminal transmembrane domain, Cry-binding region (CBR), peptidase-M1 domain, and the GAMEN and HEXXH(X)$_{18}$E zinc-binding site motifs are shown in red, grey, green, yellow, blue and orange, respectively. (B) Alignment of the Cry1Aa toxin-binding region (CBR) of *Litopenaeus vannamei*, *Bombyx mori* and *Manduca sexta* APNs. The sequences of *Lv*APN1 (XP_027215499.1); *Lv*APN2 (XP_027218958.1); *Bm*APN1 (AFK85020); *Bm*APN2 (AB011497); *Bm*APN3 (AF352574); *Bm*APN4 (AB013400) and *Ms*APN (CAA66466) were compared. Perfectly conserved amino acid residues have black backgrounds. (C) Aminopeptidase N transcript expression analysis in various *L. vannamei* tissues by RT-PCR. The tissues examined were gill (G), heart (H), hemocyte (Hcy), lymphoid (Lym), hepatopancreas (Hep), intestine (Int) and stomach (Stm). EF1-α was used as the internal reference and PCR control. Neg is a negative control. A representative data of 3 biological replicates is shown.

We also found that both *Lv*APN1 and *Lv*APN2 were constitutively expressed in all tested immune-related tissue samples from healthy shrimp including stomach, hepatopancreas and hemocytes (Fig 1C).

## *Lv*APN1 is upregulated in response to AHPND

As shown in Fig 1A, *Lv*APN2 has no transmembrane domain, and since it would therefore presumably be expressed only in soluble form rather than as a membrane-bound protein receptor, our subsequent experiments focused only on *Lv*APN1.

First, we used Western blot to confirm that both PirA$^{vp}$ and PirB$^{vp}$ toxins were present in the purified proteins extracted from Thamai strain (TM) and absent from the proteins extracted from Mahachai strain (MC) (S2 Fig). Next, in order to analyze the expression of *Lv*APN1 that was identified from the transcriptomic data, we challenged adult *L. vannamei* with either the non-AHPND causing *Vibrio parahaemolyticus* strain S02 or the virulent AHPND-causing strain 5HP as shown in panel (i), or else with the partially purified toxins or non-toxins extracted from TM strain and MC strain as shown in panel (ii), respectively. Using qRT-PCR to analyze the expression patterns of *Lv*APN1 in the stomach (Fig 2A), hepatopancreas (Fig 2B) and hemocytes (Fig 2C), we found that at 12 and 24 hpi, the expression profiles of *Lv*APN1 in the hepatopancreas were significantly upregulated after challenge with either 5HP or the partially purified VP$_{AHPND}$ toxins. Similarly, in the stomach, *Lv*APN1 expression was upregulated at 12 hpi by the partially purified VP$_{AHPND}$ toxins and at 24 hpi by 5HP. Meanwhile, in hemocytes, although *Lv*APN1 was significantly upregulated at 12 hpi after 5HP challenge and at 24 hpi after challenge with the partially purified VP$_{AHPND}$ toxins, we also observed some unexpected downregulation of *Lv*APN1 at 12 hpi (Fig 2).

## *Lv*APN1 as a putative VP$_{AHPND}$ toxin receptor

Next, to investigate the functional importance of *Lv*APN1 as a putative VP$_{AHPND}$ toxin receptor, we used an RNA interference (RNAi) approach to silence the expression of *Lv*APN1 by the injection of double-stranded RNA (dsRNA). At 24 h after injection of purified *Lv*APN1-specific dsRNA, *Lv*APN1 transcription levels were significantly reduced in stomach, hepatopancreas and hemocytes, relative to the dsGFP-injected and NaCl-injected controls (Fig 3A). The *Lv*APN1 specific dsRNA was also shown to have no effect on the expression of *Lv*APN2 (S3 Fig). Furthermore, Fig 3B further shows that silencing of *Lv*APN1 significantly increased the survival of adult *L. vannamei* that were challenged with the partially purified VP$_{AHPND}$ toxins; although none of the shrimp in the NaCl-treated and dsGFP-treated groups survived beyond 5 days, 77% of the *Lv*APN1 silenced shrimp survived the same challenge through to the end of the experiment at 7 days.

PirAB$^{vp}$ toxins released from VP$_{AHPND}$ is the major cause of AHPND symptom of hepatopancreas necrosis. So, the effect of *Lv*APN1 knockdown on the hepatopancreas morphology of shrimp challenged with the partially purified VP$_{AHPND}$ toxin was observed (Fig 3C). As expected, histological examination showed that *Lv*APN1 silencing largely prevented the characteristic AHPND clinical signs of sloughed epithelial cells and cellular disruption, whereas in the NaCl-treated and dsGFP-treated control shrimp, both sloughing of the tubule cells and hemocyte infiltration were observed. This result suggested the putative role of *Lv*APN1 as a VP$_{AHPND}$ toxin receptor.

## *Lv*APN1 knockdown reduced cell damage in toxin-challenged hemocytes

As observed earlier that *Lv*APNs were constitutively expressed in all tested immune-related tissues especially hepatopancreas and stomach that are destroyed upon VP$_{AHPND}$ infection. However, there is no evidence of damages on other tissues reported so far. It is well known that hemocyte is a major immune tissue producing various immune effectors to fight against infection. Here, we showed that the VP$_{AHPND}$ toxins significantly decreased the total hemocyte count (THC) in the NaCl-treated and dsGFP-treated shrimp at 24h (Fig 4A). By contrast, the

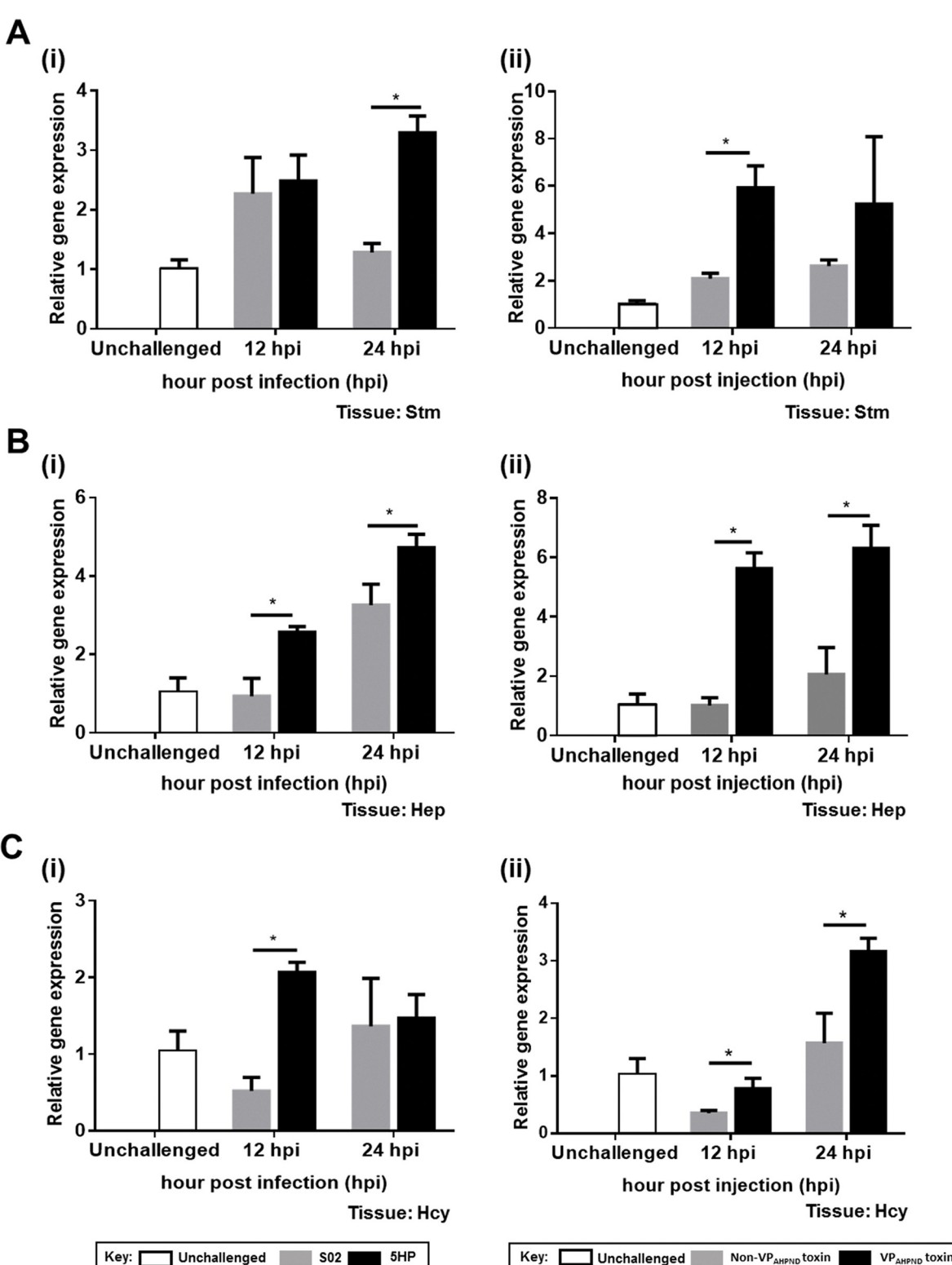

**Fig 2. Expression of the _Lv_APN1 gene upon VP_AHPND and VP_AHPND toxin challenges.** (A) Stomach, (B) hepatopancreas, and (C) hemocyte of _L. vannamei_ were collected before infection (unchallenged) and after challenge with 5HP (i) or the partially purified VP_AHPND toxins (ii) at 12 and 24 h. _Lv_APN1 expression was analyzed by qRT-PCR using EF-1α as the internal control gene. The data represent fold change of _Lv_APN1 expression in the 5HP or VP_AHPND producing strain challenged group compared to the S02 or non-VP_AHPND toxin producing strain-challenge control. The experiments were done in triplicate. Each bar represents the mean ± standard deviation (SD). Asterisks indicate significant difference ($P < 0.05$).

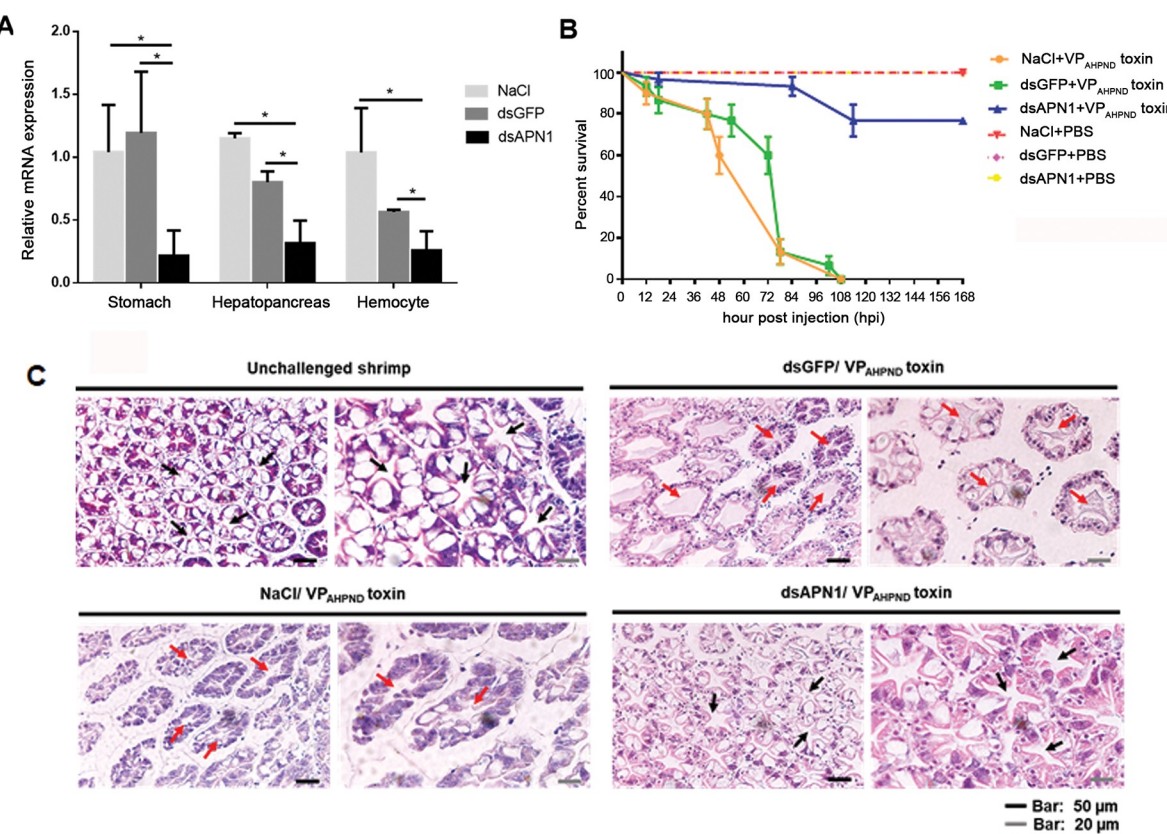

**Fig 3. The effect of *Lv*APN1 silencing in AHPND-causing bacteria pathogenesis.** (A) Confirmation of *Lv*APN1 gene knockdown in *L. vannamei*. The mRNA expression levels of the *Lv*APN1 gene in the stomach, hepatopancreas and hemocyte of shrimp injected with 0.85% NaCl, 20 μg/g shrimp of dsGFP, or 20 μg/g shrimp of dsAPN1 were determined by qRT-PCR and expressed in relative to EF-1α. Each bar represents the mean ± SD, derived from triplicate experiments. Asterisks indicate significant difference ($P < 0.05$). (B) *Lv*APN1 silencing reduced mortality in shrimp challenged with partially purified VP<sub>AHPND</sub> toxins. Cumulative mortality was monitored in 3 groups of 10 shrimp for each of the individual experimental conditions. NaCl-injected shrimp were challenged with partially purified VP<sub>AHPND</sub> toxin ( 🟡 ), dsGFP-injected shrimp were challenged with partially purified VP<sub>AHPND</sub> toxin ( 🟩 ), dsAPN1-injected shrimp were challenged with partially purified VP<sub>AHPND</sub> toxin ( 🔺 ), NaCl-injected shrimp were challenged with PBS ( 🔻 ), dsGFP-injected shrimp were challenged with PBS ( 🔶 ), dsAPN1-injected shrimp were challenged with PBS ( 🟡 ). Shrimp survival was observed every 12 h post treatment for 7 days. All experiments were performed in triplicate and the survival percentage calculated as mean ± 1 standard error (SE) at each time point as shown. (C) A representative data of haematoxylin and eosin-stained hepatopancreas collected from shrimp at 24 h after challenge with VP<sub>AHPND</sub> toxin. Normal hepatopancreatic tubules (black arrow) are observed in the unchallenged group and dsAPN1-injected group, whereas typical AHPND lesions with necrotic, sloughed epithelial cells (red arrow) were found in both the dsGFP- and NaCl-injected groups.

silencing of *Lv*APN1 results in a THC that is similar to that of the unchallenged PBS-treated group. In addition, we also found that 83% of the hemocytes in the *Lv*APN1 knockdown group were alive, compared to 63% and 58% in the NaCl and dsGFP control groups, respectively (Fig 4B).

To further determine if hemocytes are one of VP<sub>AHPND</sub> toxin target tissue, the morphology of the VP<sub>AHPND</sub>-challenged shrimp hemocytes was observed (Fig 4C). Scanning electron microscopy (SEM) showed a clear morphological change in the hemocyte cell surface after VP<sub>AHPND</sub> toxin challenge in the NaCl- and dsGFP-treated shrimp. These changes included cell disruption, pore formation on the cell surface and bursting. Meanwhile, in the *Lv*APN1 knockdown shrimp, there was no observable morphological damage to the hemocytes after VP<sub>AHPND</sub> toxin challenge. These results infer that hemocytes are a VP<sub>AHPND</sub> toxin target tissue where *Lv*APN1 is involved in penetration of toxins into cell.

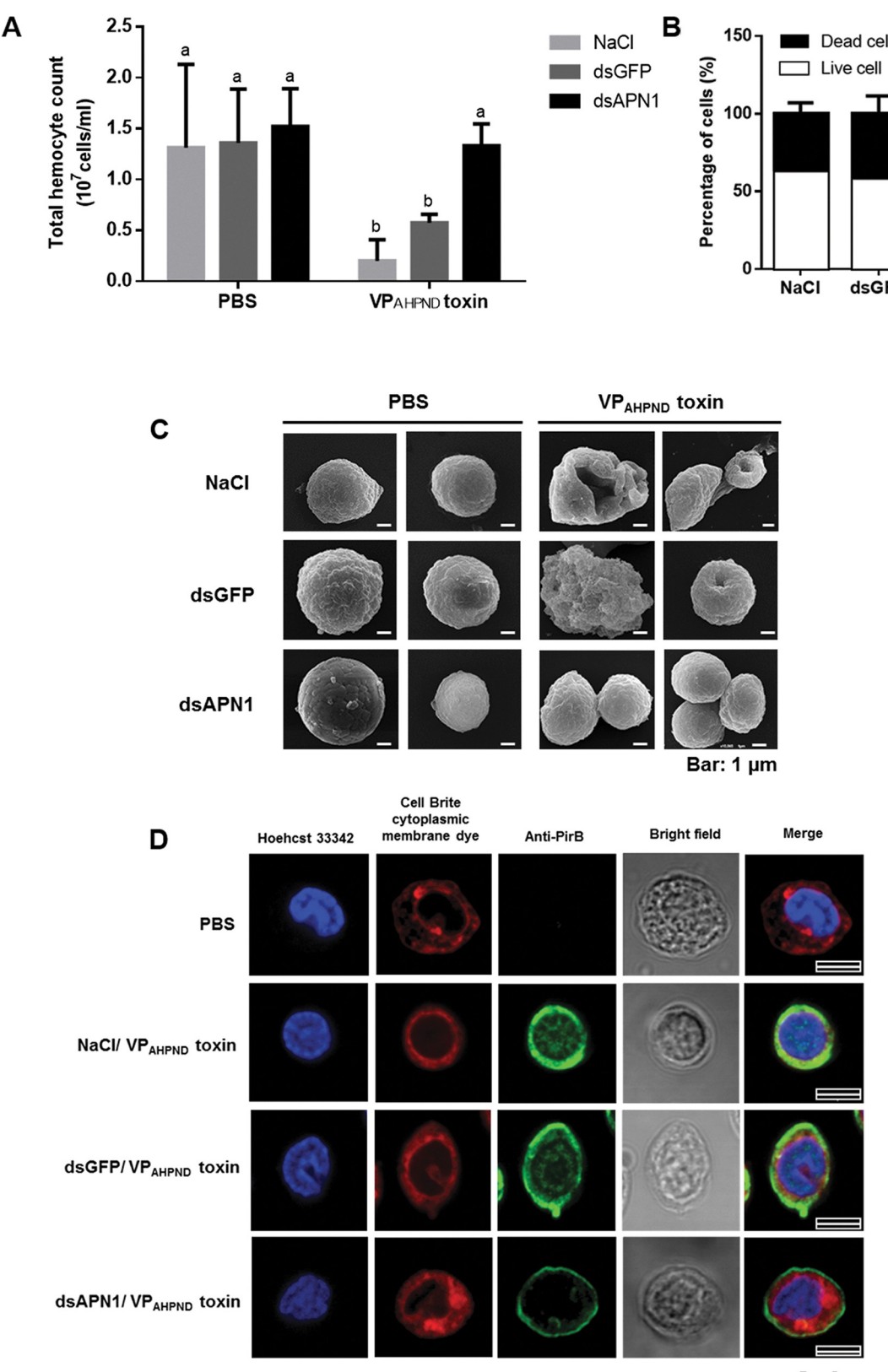

**Fig 4. The effect of *Lv*APN1 silencing on shrimp hemocyte homeostasis.** (A) Effect of *Lv*APN1 silencing on the total hemocyte count after challenge with the partially purified VP<sub>AHPND</sub> toxins. Experimental groups were the same as those used in Fig 3B. PBS was used as a

control. THC values for each treatment condition were derived from at least three shrimp. (B) Percentage of dead and viable hemoctyes in the *Lv*APN1 knockdown shrimp after partially purified VP$_{AHPND}$ toxins challenge was determined by trypan blue staining and observation under light microscopy. Asterisks indicate significant difference ($P < 0.05$). (C) The representative SEM micrograph showing morphology of *Lv*APN1 knockdown shrimp hemocytes after partially purified VP$_{AHPND}$ toxin challenge. Experimental groups were the same as those used in Fig 3B. PBS was used as a control. (D) Localization of the VP$_{AHPND}$ toxin on shrimp hemocyte by immunofluorescence. The VP$_{AHPND}$ hemocytic nuclei, cytoplasmic membrane and PirB$^{vp}$ toxin are visualized in blue (Hoechst 33342), red (CellBrite cytoplasmic membrane) and green (Alexa Fluor 488) colors, respectively. PBS-injected shrimp was used as a control. The scale bar corresponds to 5 μm. All experiments were done in triplicate.

## *Lv*APN1 plays crucial role in toxin translocation from cell membrane to cytoplasm of hemocytes

To observe the effect of *Lv*APN1 silencing on the localization of VP$_{AHPND}$ toxins on shrimp hemocytes, we used immunofluorescence and confocal microscopy in conjunction with anti-PirB$^{vp}$ antibodies specific to PirB$^{vp}$. Twenty-four h after challenge with the partially purified VP$_{AHPND}$ toxins, *Lv*APN1 knockdown shrimp hemocytes were collected, fixed and processed for the detection of PirB$^{vp}$ proteins. Nuclei were stained with Hoechst 33342 (blue) and cell membranes were stained with Cell Brite cytoplasmic membrane dye (red), while the VP$_{AHPND}$ toxin was visualized with Alexa Fluor 488 conjugated anti-PirB$^{vp}$ antibody (green). The fluorescent microscopic images revealed that while VP$_{AHPND}$ toxin was localized on both the hemocyte cell surface and the cytoplasm of the NaCl-treated and dsGFP-treated hemocytes, the VP$_{AHPND}$ toxin was only localized on the cell membrane not inside the cell of dsAPN1-treated shrimp (Fig 4D). These results suggest that in hemocytes, *Lv*APN1 is acting as a target receptor molecule of VP$_{AHPND}$ toxins and that it is a critical part of the mechanism that allows the toxins to pass through the cell membrane.

## *Lv*APN1 gene silencing reduces the number of AHPND virulence plasmids in stomach, and prevents the appearance of gross clinical signs in hepatopancreas

To investigate the effect of *Lv*APN1 silencing in the stomach of *L. vannamei*, we performed an immersion challenge using two strains of *V. parahaemolyticus*, S02 and 5HP. At 24 h after infection, a PCR-based AHPND detection kit was used to test stomach samples for the presence of two sequences in the pVA1 plasmid, one that overlaps both of the Pir toxin genes, and another more stable sequence known as AP2 [19]. As shown in both panels of Fig 5 after 5HP infection, the dsAPN1-injected group showed significantly lower counts for the pVA1 plasmids than the NaCl-treated and dsGFP-treated controls. All of the above results provide further evidence of *Lv*APN1 involvement in AHPND pathogenesis.

## ELISA assay of protein-protein interactions between *Lv*APN1 and the PirA$^{vp}$ and PirB$^{vp}$ toxins

After confirming the successful expression of the recombinant proteins truncated *Lv*APN1, PirA$^{vp}$-His, PirB$^{vp}$-GST and GST in a bacterial expression system (S4 Fig), the purified truncated r*Lv*APN1 was found to bind directly to rPirA$^{vp}$ and rPirB$^{vp}$ in a concentration-dependent manner (Fig 6). Assuming a one-site binding model, the apparent dissociation constants ($K_d$) of truncated r*Lv*APN1 to rPirA$^{vp}$ and rPirB$^{vp}$, as calculated from the saturation curves, were 3.2 μM and 0.5 μM, respectively. These results suggest that both VP$_{AHPND}$ binary toxin subunits can bind directly to *Lv*APN1 receptor.

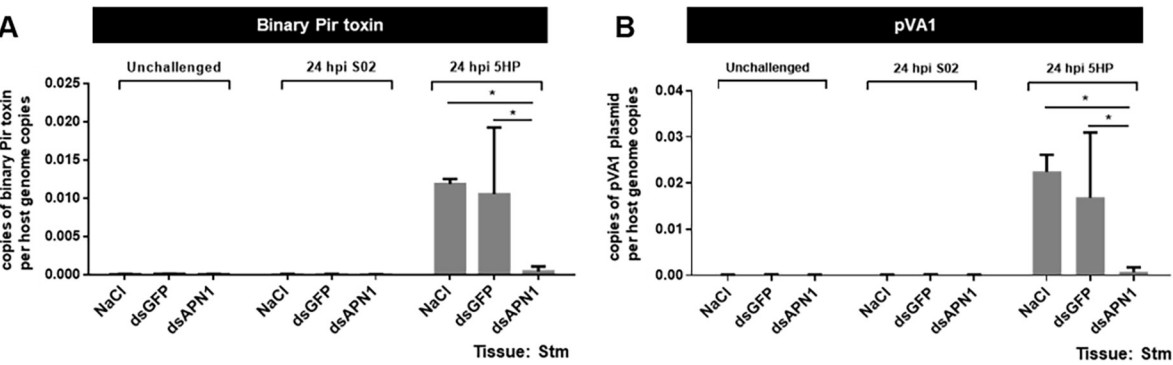

**Fig 5. Reducing of AHPND-causing bacteria plasmid in stomach of *Lv*APN1 silenced shrimp.** PCR amplification of two different sequences in the genome of the pVA1 plasmid, (A) Binary Pir toxin sequence and (B) pVA1 sequence from the stomach of shrimp injected with 0.85% NaCl, 20 μg/g shrimp of dsGFP, or 20 μg/g shrimp of dsAPN1. Shrimp stomach were collected after dsRNA injection (unchallenged) and after infection with the AHPND-causing *V. parahaemolyticus* 5HP strain and non-AHPND causing *V. parahaemolyticus* S02 at 24 h. The data represent copies of pVA1 plasmid per host genome copies. Each bar represents the mean ± standard deviation (SD) of triplicate experiments. Asterisks indicate significant difference ($P < 0.05$).

## Discussion

APNs in insects have been extensively investigated for their interactions with *Bacillus thuringiensis* Cry toxins [20]. In the present study, we first identified aminopeptidase N 1 and 2 (*Lv*APN1 and *Lv*APN2) from the transcriptome of VP_AHPND-challenged *L. vannamei* hemocytes. *Lv*APN1 and *Lv*APN2, which were expressed in all immune-related and AHPND-affected tissues including stomach, hepatopancreas and hemocytes (Fig 1C), possess the hallmark characteristics of lepidopteran APNs, so they appear to belong to the APN family (Fig 1A). There are at least four classes of APN in the insect midgut, where they occur either as soluble enzymes or in association with the microvillar membrane [15]. While these four APNs are thought to differ in their amino acid specificity, they all function to cleave N-terminal amino acids from peptides, a step which is necessary for amino acid co-transport into

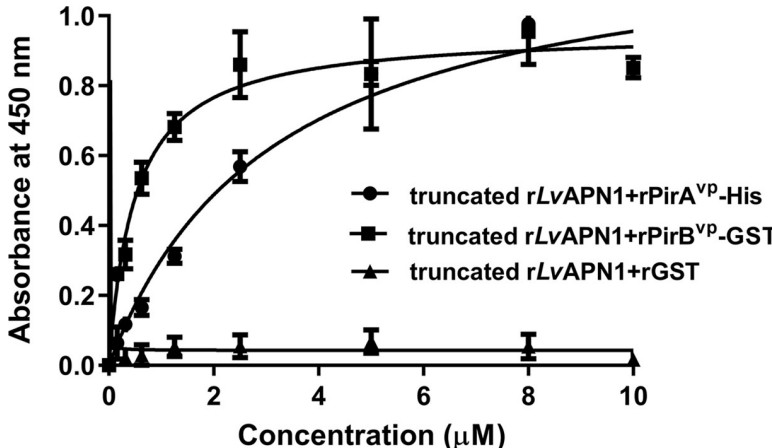

**Fig 6. Binding ability of rPirA^vp and rPirB^vp on immobilized recombinant truncated *Lv*APN1-His determined by ELISA.** The purified rPirA^vp-His, rPirB^vp-GST or rGST (0–10 μM) was added to a purified recombinant truncated *Lv*APN1-coated plate, followed by probing with the anti-PirB^vp specific primary antibody and goat anti-rabbit-conjugated HRP secondary antibody. Finally, after addition of the tetramethylbenzidine (TMB) substrate, the absorbance at 450 nm (A_450) was measured. Solid lines illustrate the fitted curves. The data are shown as the mean ± 1 Standard error of mean (SEM), derived from triplicate experiments.

epithelial cells [21]. Prediction of a transmembrane helix showed that *Lv*APN1 is potentially a membrane-bound protein (Figs 1A and S1) while *Lv*APN2 is not. The results of sequence alignment indicated that *Lv*APN1 CBR shared moderate to high protein identity with the CBRs of other APN homologs, while the protein sequences beyond this consensus domain diverged considerably from those of the APNs found in Cry-susceptible insects (Fig 1B) [22]. Nevertheless, based on the similarity of its domain structure, it seems likely that *Lv*APN1 would have a similar function to the APN receptors in insects.

Specific binding of the Cry toxins to receptors on the epithelial cells of the midgut and hindgut is critical for their toxic effect on susceptible insects [23–25], and previous study has shown that the *Bm*APNs were specifically or highly expressed in the midgut of *B. mori* after *B. bombysepticus* and *B. thuringenesis* infection [26]. Here, we found that the expression levels of the *Lv*APN1 gene were increased after challenge with either the AHPND-causing bacteria or the VP$_{AHPND}$ toxin not only in stomach and hepatopancreas, but also in hemocytes (Fig 2). Interestingly, there are other reports in arthropods that their hemocytes are targeted by Cry toxins. For example, Cerstiaens *et al.*, (2001) [27] demonstrated the toxicity of Cry toxins to the hemocoel in *Lymantria dispar* (Lepidoptera) and *Neobellieria bullata* (Diptera). In addition, the *Aj*APN1 protein of non-gut hemocoelic tissues was implicated as a Cry1Aa toxin receptor in these tissues in *Achaea Janata* [28]. All of these findings suggest that VP$_{AHPND}$ toxins targeting stomach, hepatopancreas, and hemocytes mediated by *Lv*APN1 is required for AHPND pathogenesis in shrimp.

Silencing of *Ha*APN1 in *Helicoverpa armigera* was found to decrease the susceptibility of larvae to Cry1Ac toxins [29], while silencing of *Hc*APN3 was also associated with reduced susceptibility of *Hyphantria cunea* to Cry1Ab [30]. In the present study we likewise investigated the function of *Lv*APN1 by dsRNA-mediated silencing. Our results show that *Lv*APN1 dsRNA significantly silenced *Lv*APN1 in the stomach, hepatopancreas and hemocytes (Fig 3A) and knockdown of *Lv*APN1 reduced the mortality of VP$_{AHPND}$ toxins-challenged shrimp (Fig 3B). Recent research demonstrated that in the germ-free brine shrimp PirAB$^{vp}$ toxins bind to epithelial cell of the digestive tract and damage enterocytes in the midgut and hindgut regions [18]. In the Penaeid shrimp, it is known that VP$_{AHPND}$ toxins cause the damage on stomach and hepatopancreas tissues whereas no evidence is available for hemocytes. In this study we would like to prove that not only stomach and hepatopancreas but also hemocytes are target tissue of VP$_{AHPND}$ toxins. We investigated the effect of *Lv*APN1 silencing on availability and morphology of hemocyte in VP$_{AHPND}$ toxin-challenged shrimp. We found that VP$_{AHPND}$ toxins cause severe damage of hemocyte leading to lowering total hemocyte number and cell death while silencing of *Lv*APN1 prevented these effects (Fig 4A and 4B) emphasizing the participation of *Lv*APN1 in VP$_{AHPND}$ toxin susceptibility of hemocyte. Immunofluorescence assay further revealed that the VP$_{AHPND}$ toxins passed through the cell membrane and localized inside the cell, but when *Lv*APN1 was silenced, the VP$_{AHPND}$ toxin was unable to gain entry and remained localized on the cell membrane (Fig 4D). Although these effects were observed in circulating hemocytes, the VP$_{AHPND}$ toxins presumably damages the hemocytes in the hepatopancreas in the same way. If so, then the hemocyte infiltration that occurs in the late stage of infection would include hemocytes that have already been damaged by the VP$_{AHPND}$ toxins [31].

In insects, binding of the Cry toxins to receptors leads to membrane insertion into the host cell [14,28,31]. Thus, for example Cry toxins form pores in the apical membrane of larvae midgut cells, destroying the cells and killing the larvae [11]. Our results showed that while challenge with the partially purified VP$_{AHPND}$ toxins led to cell death and pore formation in hemocytes in unsilenced shrimp, knockdown of *Lv*APN1 not only inhibited pore formation by the VP$_{AHPND}$ toxins (Fig 4C), it also reduced both shrimp mortality (Fig 3B), and led to a

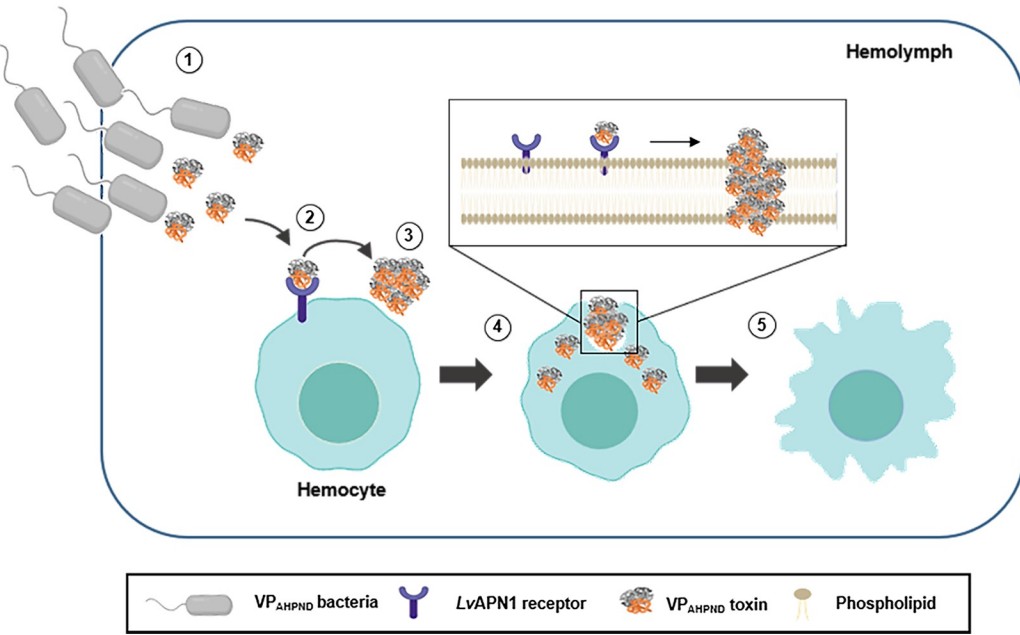

**Fig 7. Schematic representation of AHPND pathogenesis showing the proposed role of *Lv*APN1 in hemocyte.** (1) VP$_{AHPND}$ bacteria enter the shrimp and release PirAB$^{vp}$ toxin. (2) The binary VP$_{AHPND}$ toxins bind to *Lv*APN1 receptor which is embeded in cell membrane of shrimp hemocyte. (3) This interaction might induce VP$_{AHPND}$ toxins oligomerization. (4) VP$_{AHPND}$ toxins induce pore formation and insert into membrane. (5) Hemocyte is lyzed and damaged causing by pore formation and membrane insertion.

reduction in the number of AHPND-causing bacteria in the stomach (Fig 5). The reason for reduction is still unclear, but one possibility is that the lack of *Lv*APN1 receptors might somehow reduce susceptibility of hemocyte, stomach, and hepatopancreas cells to VP$_{AHPND}$ toxins.

Binding specificity between the receptor and the toxin is also critically important for Cry toxicity. In a previous report, ligand blotting showed that in *B. mori* and *H. cunea*, the APN receptors specifically bind to Cry1Aa toxins [32]. Similarly, here we used ELISA to determine the binding of VP$_{AHPND}$ toxins and *Lv*APN1 receptor. We found that the recombinant His-tagged N-terminal of *Lv*APN1 was able to directly bind to both the rPirA$^{vp}$ and rPirB$^{vp}$ toxins, and that it showed a slightly higher binding affinity to PirB$^{vp}$ than to PirA$^{vp}$ (Fig 6), suggesting that PirB$^{vp}$ is probably responsible for the binding to *Lv*APN1 receptor on hemocytes. The role of PirB$^{vp}$ as a ligand for cell surface receptor of shrimp target cells has been recently suggested as well. Victorio-De Los Santos et al [33] suggested that PirB$^{vp}$ subunit is a lectin that can bind to amino sugar ligand and exhibits the hemagglutinating activity as the PirAB$^{vp}$ complex [33]. Taken together, our data suggests that *Lv*APN1 is likely to facilitate AHPND pathogenesis by functioning as a receptor for VP$_{AHPND}$ toxins, and that it is therefore likely to be involved in pore-formation and membrane insertion.

Lastly, we note that the ability of the VP$_{AHPND}$ toxins to enter and damage hemocytes (Fig 4) suggests that the pathological effects of AHPND are not limited to the stomach and hepatopancreas, but also extend to the hemocytes. Taking all of the results presented here we propose a model for the role of *Lv*APN1 in shrimp hemocytes (Fig 7). According to this model, the AHPND-causing *V. parahaemolyticus* enter into the shrimp. At the same time, the AHPND-causing *V. parahaemolyticus* release of PirAB$^{vp}$ toxins. Next, the PirAB$^{vp}$ toxins bind to *Lv*APN1 receptor located on cell membrane of shrimp hemocytes and then might induce

PirAB$^{vp}$ toxin oligomerization. This interaction enhanced pore formation and membrane insertion which led to the hemocyte morphology changes and hemocyte lysis.

## Materials and methods

### Ethics statement

The experiments involving animals received ethical approval from Chulalongkorn University Animal Care and Use Committee (protocol review No. 1923019). The biosafety concerns of experiments performed was approved by the Institutional Biosafety Committee of Chulalongkorn University (SCCU-IBC-008/2019).

### DNA sequence analysis

$L.$ $vannamei$ APN genes ($Lv$APN1 and $Lv$APN2) were found in our transcriptomic database of VP$_{AHPND}$-challenged $L.$ $vannamei$ hemocytes [18]. The full-length sequences of $Lv$APN1 was retrieved from the in-house transcriptomic database and deposited into GenBank database (accession number MW259048) whereas that of $Lv$APN2 was retrieved from GenBank database (accession number XP_027218958). The amino acid sequence alignment was performed using the aminopeptidase N of various species from previous reports [34]. The amino acid sequences of the two $Lv$APN proteins were analyzed for conserved motifs by SMART (http://smart.embl-heidelberg.de) scanning. Prediction of a signal peptide at the N-terminus of each protein was conducted with SignalP 4.1 [35]. A GPI anchor signal at the C-terminus was predicted using PredGPI, FragAnchor, and big-PI Predictor. $N$-glycosylation and $O$-glycosylation sites were predicted by the NetNglyc 1.0 Server [36] and NetOglyc 4.0 Server [37], respectively. Transmembrane helices of $Lv$APN amino acids were predicted by the TMHMM 2.0 Server (http://www.cbs.dtu.dk/services/TMHMM/).

### Tissue distribution analysis

Tissue distribution for the $Lv$APN1 and $Lv$APN2 genes was analyzed in 7 tissues (gill, heart, hemocytes, lymphoid organ, hepatopancreas, intestine, stomach) from three individuals. Total RNA was isolated from these tissues using GENEzol reagent (Geneaid). RevertAid First Strand cDNA Synthesis Kit (Thermo Scientific) was used for reverse transcription. Gene expression analysis was done by semi-quantitative RT-PCR using specific primers for $Lv$APN1, $Lv$APN2 and EF-1α. The sequences for all primer sets are listed in Table 1. Among three biological replicates, a representative result is shown.

### Bacterial strains

For VP$_{AHPND}$ challenge by immersion, $V.$ $parahaemolyticus$ strains 5HP (AHPND-causing strain) and S02 (non-AHPND-causing strain) were used. In case of VP$_{AHPND}$ toxin challenge, the VP$_{AHPND}$ strain TM isolated from a shrimp culture farm in Thamai, Chanthaburi province, Thailand, was used for VP$_{AHPND}$ toxin purification. The non-AHPND causing strain, the non-VP$_{AHPND}$ strain MC isolated from a shrimp culture farm in Mahachai, Samut Sakhon province, Thailand, was used for non-VP$_{AHPND}$ toxin purification. All $V.$ $parahaemolyticus$ strains were cultured and maintained on thiosulfate citrate bile salts sucrose medium (TCBS) (BD Biosciences) [7,8].

### Preparation of the partially purified VP$_{AHPND}$ toxin

To prepare the partially purified VP$_{AHPND}$ toxins and non-VP$_{AHPND}$ toxins, bacterial suspension of either the AHPND-causing (TM) or non-AHPND-causing strains (MC) of $V.$ $parahaemolyticus$, was cultured in tryptic soy broth (TSB) supplemented with 1.5% NaCl incubated at

**Table 1. Primers used in this study.**

| Usage | Gene | Primer name | Primer sequence (5′-3′) |
|---|---|---|---|
| Real-time PCR/ PCR | *Lv*APN1 | *Lv*APN1-F | GGGCAAGGAGGTCAAATGGA |
| | | *Lv*APN1-R | CAACGCCACTGTTGCATTGA |
| | *Lv*APN2 | *Lv*APN2-F | GACGTCACGACCTCGGCTG |
| | | *Lv*APN2-R | GCCAGGTACCTTGTCTTCCC |
| | *Lv*EF-1α | EF-1α-F | CGCAAGAGCGACAACTATGA |
| | | EF-1α-R | TGGCTTCAGGATACCAGTCT |
| dsRNA synthesis | *Lv*APN1 | dsAPN1_knd_F_XbaI | CATTCTAGAAGAGAAAAGGTATATCGCTACCACC |
| | | dsAPN1_knd_R_NcoI | CATCCATGGCTACAAGGTATGTGCTCATTTCCAC |
| | | dsAPN1_knd_F_BamHI | CATGGATCCAGAGAAAAGGTATATCGCTACCACC |
| | | dsAPN1_knd_R_NdeI | TCTCATATGCTACAAGGTATGTGCTCATTTCCAC |
| | EGFP | dsEGFP_knd_F_XbaI | CATTCTAGAATCATGGCCGACAAGCAGAA |
| | | dsEGFP_knd_R_NcoI | CATCCATGGAACTCCAGCAGGACCATGTG |
| | | dsEGFP_knd_F_BamHI | CATGGATCCATCATGGCCGACAAGCAGAA |
| | | dsEGFP_knd_R_NdeI | TCTCATATGAACTCCAGCAGGACCATGTG |
| Recombinant protein expression | *Lv*APN1 | APN1_CBR_BamHI-F | CGTCAGGATCCGGATGAGATTTTACGTCGAGGAAG |
| | | APN_CBR_SalI-R | AGACGTCGACGATTACTACTACTACTACTACCACGAGTCCATCCTCCAAGC |

APN, aminopeptidase N, EF, Elongation factor, knd, knockdown, EGFP, enhanced green florescent protein, CBR, Cry toxin binding region.

30˚C with shaking for 16 h. Subsequently, the bacterial culture was inoculated 1:100 in 400 ml TSB and cultivation continued with shaking for approximately 2–3 h until the OD$_{600}$ reached 2 (equivalent to $10^8$ colony forming unit; CFU per ml). After centrifugation at 8,000 ×g for 15 min at 4˚C, the supernatant was collected and subjected to ammonium sulfate precipitation (AS) as described by Sirikharin et al. (2015) [38]. Total protein concentration of the partially purified VP$_{AHPND}$ toxins derived from TM strain and non-VP$_{AHPND}$ toxins derived from MC strain was determined using Bradford reagent (Bio-Rad) and stored at -80˚C. The protein quality was analyzed by SDS-PAGE and Western blot analysis using specific antibodies to Pir-A$^{vp}$ and PirB$^{vp}$ proteins.

## Challenge with VP$_{AHPND}$ bacteria and the partially purified VP$_{AHPND}$ toxins

After acclimatization, shrimp were challenged with *V. parahaemolyticus* strain 5HP and S02 by immersion in tanks that were inoculated with a bacterial suspension as described by Boonchuen et al., (2018) [39]. The median lethal dose (LD$_{50}$) of the 5HP bacterial inoculant at 24 h was determined in 10 shrimp, and this resulted in a final bacterial concentration of $2.5 \times 10^5$ CFU/ml in the immersion tanks. The LD$_{50}$ at 24 h for the partially purified VP$_{AHPND}$ toxins was similarly determined as 0.25 μg/g shrimp. The VP$_{AHPND}$ toxins were diluted in 1×PBS mixed with a red food-grade dye and intramuscular injected into shrimp. The red food-grade dye was used to make sure that VP$_{AHPND}$ toxins was properly entered into the shrimp muscle. The effect of all the above challenges was also confirmed by observation of morphological changes in the hepatopancreas such as paling and atrophy as well as lethargy in shrimp.

## Quantitative real-time RT-PCR (qRT-PCR) analysis of *Lv*APN1 expression after challenge with AHPND-causing bacteria and the partially purified VP$_{AHPND}$ toxins

Shrimp (n = 12 per group) were challenged with *V. parahaemolyticus* strain 5HP or VP$_{AHPND}$ toxins as described above, and the stomach, hepatopancreas and hemocytes were collected

from three shrimp in each experimental group at either unchallenge, 12 or 24 h post challenge. Total RNA was isolated, and cDNA was synthesized using REzol C&T reagent (Protech Technology Enterprise Co., Ltd.) as per the manufacturer's protocol and a RevertAid First Strand cDNA Synthesis Kit (Thermo Scientific) was used to synthesize the cDNA. The cDNA samples were used as templates in the qRT-PCR reaction. The S02-infected and non-VP$_{AHPND}$ toxin-injected shrimp were used as controls.

To determine the expression level of *Lv*APN1 gene and EF-1α internal control gene, the qRT-PCR analysis was performed using an appropriate amount of cDNA for each gene, specific oligonucleotide primers (Table 1) and qPCR Master Mix (KAPA Biosystem) in CFX96 Touch RealTime PCR System (Bio-Rad) under the following conditions: 95˚C for 3 min, 40 cycles at 95˚C for 30 s, 60˚C for 30 s, and 72˚C for 30 s. The experiments were done in three biological replicates. Relative expression level was calculated using the mathematical model of Pfaffl (2001) [40]. Data were analyzed using one-way ANOVA followed by Duncan's new multiple ranges test and presented as means ± standard deviations (SD). The level of statistical significance was set at $P$-value < 0.05

### dsRNA silencing of *Lv*APN1 expression

To investigate the functional importance of *Lv*APN1 as a putative VP$_{AHPND}$ toxin receptor, we used an RNA interference (RNAi) approach to silence the expression of *Lv*APN1 by the injection of double-stranded RNA (dsRNA). For dsRNA production using bacteria system, the recombinant plasmid pET-19b containing a 230 bp sense-antisense construct targeted to *Lv*APN1 mRNA was transformed into the ribonuclease III-deficient *Escherichia coli* strain HT115 (DE3) [37] to produce dsRNA-*Lv*APN1 (dsAPN1) and the recombinant plasmid pET-19b containing a 400 bp sense-antisense construct targeted to EGFP mRNA was used to produce dsRNA-GFP (dsGFP), a dsRNA control. The dsRNAs were expressed and extracted as by described in Posiri *et al*. (2013) [41]. 20 µg of dsAPN1, dsGFP or 0.85% NaCl were injected into each shrimp (~5 g body weight), and 24 h later, the shrimp were injected with 0.25 µg/g shrimp of VP$_{AHPND}$ toxins or 1×PBS pH 7.4 for the control. The stomach, hepatopancreas and hemocytes were collected at 24 h post VP$_{AHPND}$ toxin injection from three shrimp in each experimental group. Total RNA was isolated and cDNA was synthesized. Also, the gene expression level of *Lv*APN1 and EF-1α, internal control gene, was analyzed by the qRT-PCR as described above to verify the efficiency of dsRNA silencing.

### Susceptibility of VP$_{AHPND}$ toxin-challenged *L. vannamei* after *Lv*APN1 silencing

Shrimp (3 groups of 10 individuals) were injected intramuscularly with dsAPN1, dsGFP or 0.85% NaCl, respectively, in the lateral area of the fourth abdominal segment. Twenty-four h later, the treated shrimp were injected with 50 µL of 0.25 µg/g shrimp of VP$_{AHPND}$ toxins using a syringe with a 29-gauge needle. In the corresponding control groups (3 groups of 10 individuals), the shrimp were injected with 1×PBS pH 7.4 instead of VP$_{AHPND}$ toxins. After the last injection, the shrimp mortality was monitored twice daily for 7 days. The experiments were done in three replicates.

### Effect of *Lv*APN1 silencing on the hepatopancreas damage caused by VP$_{AHPND}$ toxin

Individual live shrimp samples (n = 3) were also taken from each group dsAPN1-, dsGFP-, and NaCl-injected shrimp at 0, 3, 12, and 24 h post the partially purified VP$_{AHPND}$ toxin

injection. Individual shrimp were fixed with Davison's fixative 48 h and subsequently stored in ethanol as described by Lightner (1996) [1]. The cephalothorax was then sectioned and stained with H&E stain following standard histological methods. Hepatopancreatic structures were examined using light microscopy, and necrosis of hepatopancreas tubules and hemocytic infiltration in the hepatopancreas were evaluated. Among 3 individuals examined, the representative result is shown.

## Total Hemocyte Count (THC)

Prior to counting, the pooled samples of the hemolymph/anticoagulant mixture of each experimental knockdown group (NaCl, dsGFP and dsAPN1) after 24 h-VP$_{AHPND}$ toxin challenge that were set aside from the immunofluorescence assay were kept on ice. To conduct the total hemocyte count, 4% (w/v) paraformaldehyde was added at the ratio 1:1 to immobilize the hemocytes, and the hemocytes were then counted using hemocytometer and a phase contrast microscope (Nikon, Japan). The observed number of hemocytes on the hemocytometer plate, the total hemocyte count in 1 mL hemolymph was then calculated. To determine the proportion of dead vs alive hemocytes, the cells were stained with 0.4% trypan blue prior to immobilization to distinguish between viable and non-viable cells. The experiments were done in three replicates.

## Effect of *Lv*APN1 silencing on the morphology of VP$_{AHPND}$ toxin-challenged shrimp hemocytes by SEM

To investigate the direct effect of the partially purified VP$_{AHPND}$ toxins on hemocyte morphology in shrimp either with or without *Lv*APN1 silencing, 20 μg of dsAPN1, dsGFP, or NaCl were injected into each shrimp (~5 g body weight), and 24 h later, the shrimp were injected with 0.25 μg/g shrimp of VP$_{AHPND}$ toxins or 1×PBS, pH 7.4 for the control. At 24 h post challenge, shrimp hemolymph was collected using a sterile 1-ml syringe with anticoagulant. The hemocytes were collected, fixed in 2% glutaraldehyde and the hemocyte morphology was then observed under scanning electron microscope. The experiments were done in triplicate and the representative result is shown.

## Determination of partially purified VP$_{AHPND}$ toxins localization on shrimp hemocytes by immunofluorescence

To investigate the localization of the VP$_{AHPND}$ toxins on hemocytes, three individual shrimp in each experimental knockdown group (NaCl, dsGFP, and dsAPN1) were challenged with VP$_{AHPND}$ toxins, and after 24 h, approximately 1.0 ml of *L. vannamei* hemolymph was collected from each shrimp under equal volume of an anticoagulant. After seeding aside some of the pooled hemolymph for the THC assay (see above), the hemocytes in these samples were isolated by centrifugation (800×g; 10 min; 4˚C), with the hemocyte pellet being immediately fixed in 4% (w/v) paraformaldehyde in 1×PBS pH 7.4 at ratio of 1:1 and kept at 4˚C until use. Cells ($5\times10^5$ cells/ml) were fixed onto the poly-L-lysine (Sigma) coated-coverslips in a 24-well plate and washed three times with 0.02% Triton X-100 in 1×PBS pH 7.4 followed by cell membrane staining with Cell Brite cytoplasmic membrane dye (Biotium) at the ratio of 1:200 in blocking solution (0.02% Triton X-100, 10% FBS and 1% BSA in 1×PBS pH 7.4) for 10 min. Cells were washed three times and permeabilized with 1×PBS pH 7.4 containing 0.2% gelatin, 1% BSA and 0.02%Triton X-100 for 30 min. Cells were then washed three times and blocked with 1× PBS pH 7.4 containing 10% FBS and 0.02% Triton X-100 for 2 h. Cells were washed again and probed with the rabbit anti-PirB$^{vp}$ primary antibody at ratio of 1:5 000 in 1×PBS pH

7.4 containing 10% FBS and 0.02% Triton X-100 at 4°C overnight, followed by washing and incubation with anti-rabbit secondary antibody conjugated with Alexa Fluor 488 (ratio 1:5 000) at room temperature for 2 h. The nuclei were stained with Hoechst 33342 (Thermo Scientific). The coverslips were mounted with EverBrite Mounting Medium (Biotium) and sealed on glass slides. Fluorescence images were detected with an LSM 800 laser scanning confocal microscope (Carl Zeiss). The experiments were done in triplicate and the representative result is shown.

## Effect of *Lv*APN1 silencing on stomach colonization by AHPND-causing bacteria

At 24 h after *Lv*APN1 knockdown, shrimp stomach was collected before (unchallenge) and after challenge with the 5HP (AHPND-causing) or S02 (non-AHPND causing) strains of *V. parahaemolyticus*, stomach samples were collected for DNA extraction using a DTAB/CTAB DNA extraction kit (GeneReach Biotechnology Corp.). The DNA isolated from shrimp stomach was screened for the presence of both the Toxin 1 sequence (which includes parts of both PirA^vp^ and PirB^vp^ genes) and the part of pVA1 sequence (which lies elsewhere on the pVA1 AHPND plasmid) using an IQ REAL AHPND/EMS Quantitative System on a CFX96 real-time machine (Bio-Rad) according to the supplier's instructions. The IQ REAL system included artificial DNA that contained partial sequences of the PirAB^vp^ gene (Toxin 1) and the part of pVA1 region, which were used as standards to obtain standard curves. Copy numbers of the binary Pir toxin sequence and the pVA1 sequence were normalized against the number of host genome copies as measured by an IQ REAL WSSV Quantitative System (Gene Reach Biotechnology Corp). The data are represented as the mean±SD of triplicate tests.

## ELISA assay of interaction between recombinant truncated r*Lv*APN1 protein and the recombinant PirA^vp^ and PirB^vp^ toxins

To investigate the direct binding affinity between the PirA^vp^ and PirB^vp^ toxins subunits and *Lv*APN1, the recombinant protein of PirA^vp^-His, PirB^vp^-GST, GST, and truncated *Lv*APN1--His (r*Lv*APN1) was prepared. The recombinant PirA^vp^-His and PirB^vp^-GST proteins were produced from *E. coli* BL-21 CodonPlus (DE3) clones harbouring plasmids to express the tagged PirA^vp^ or PirB^vp^ proteins; these were kindly provided by Dr. Kallaya Dangtip, Center of Excellence for Shrimp Molecular Biology and Biotechnology (CENTEX SHRIMP, Mahidol University, Thailand). The recombinant His tagged rPirA^vp^ and the GST-tagged rPirB^vp^ were overproduced and then purified using Ni-NTA and Glutathione-beads, respectively [22]. The recombinant truncated *Lv*APN1-His is N-terminal region of *Lv*APN1 composing of CBR and the active site of peptidase-M1 domain fused with 6×-His tag. The *E. coli* BL-21 (DE3) clone harbouring pET-28b-His-truncated *Lv*APN1was constructed by cloning the *Lv*APN1 fragment encoding for 388 amino acids truncated *Lv*APN1 containing a CBR (^205^Aspatic acid to ^591^Valine) to pET-28b-His vector. It was further expressed in LB broth at 37°C and induced with 1 mM IPTG at 30°C for 3 h. The crude r*Lv*APN1 was then purified by Ni-NTA. The purified truncated r*Lv*APN1-His, rPirA^vp^-His, rPirB^vp^-GST and rGST were then dialyzed against 1×PBS, pH 7.4. Western blots using anti-His (Bioman Scientific) and anti-GST (Cell Signaling Technology) antibodies were used to confirm the expression of the respective proteins.

The interaction of toxin proteins (rPirA^vp^ and rPirB^vp^) and purified truncated r*Lv*APN1 was determined by ELISA technique as described by Boonchuen et al. (2018) [24] with modifications. Briefly, 20 μg of the purified truncated r*Lv*APN1 was coated overnight with 0.1 M carbonate/bicarbonate buffer pH 9.6 in 96-well microtiter plate at 4°C. The wells were washed 3 times with TBS (20 mM Tris-HCl, 150 mM NaCl; pH 8.0) containing 0.1% (v/v) Tween 20

(TBST) for 15 min at room temperature. After that, the coated truncated *Lv*APN1 was blocked with TBS, pH 8.0 containing 0.5% (w/v) BSA (Sigma Aldrich) for 3 h. After washing 3 times with TBST, 100 μl of either rPirA$^{vp}$, rPirB$^{vp}$ or rGST at various concentration (0–10 μM in TBS) were incubated with the truncated r*Lv*APN1-coated 96-well plate and incubated at 4˚C overnight. The wells were then washed with TBST. The bound proteins (rPirA$^{vp}$ and rPirB$^{vp}$) were detected using 1: 5 000 specific primary antibodies; (rabbit anti-PirA$^{vp}$ polyclonal antibody and rabbit anti-PirB$^{vp}$ polyclonal antibody, respectively) and a secondary antibody, HRP-conjugated goat anti-rabbit antibody, diluted 1:5 000 fold. The HRP substrate was added and A$_{450}$ was measured by spectrophotometry. The specific antibodies for PirA$^{vp}$ and PirB$^{vp}$ were gifts from Professor Dr. Chu Fang Lo, Department of Biotechnology and Bioindustry Sciences, National Cheng Kung University, Tainan, Taiwan. All assays were performed in triplicate. rGST was used instead of rPirA$^{vp}$ and rPirB$^{vp}$ for the negative control, which was detected by anti-GST antibody. GraphPad Prism 6 software (Graph-Pad Software, Inc.), was used to analyzed and graphically present the data. The data were fitted to a one-site binding-specific binding model and the K$_d$ value was determined from the nonlinear curve fitting as $A = A_{max} [L]/(Kd + [L])$, where A is the absorbance, which is proportional to the bound concentration, A$_{max}$ is the maximum binding, and [L] is the concentration of the recombinant proteins. The data are represented as the mean±SD of triplicate tests.

## Supporting information

**S1 Fig. Characterization of the putative *Lv*APN1 protein.** The putative N-terminal transmembrane domain is boxed. The GAMEN and HEXXH(X)$_{18}$E zinc-binding site motifs are shown in bold with a grey background. The predicted Cry toxin binding region is highlight in yellow. Two conserved domains, the peptidase M1 domain and ERAP1-like C-terminal domain, are underlined and dash-underlined, respectively. Putative *N*-glycosylated asparagine residues predicted by the NetNGlyc 1.0 server, and putative *O*-glycosylated threonines and serine residues predicted by the NetOGlyc 3.1 server are shown in black.
(TIF)

**S2 Fig. Partially purification of VP_AHPND toxin.** (A) SDS-PAGE analysis and (B) Western blot analysis with PirAB$^{vp}$ polyclonal antibodies of the partially purified ammonium sulfate fractions from the culture medium of the non-AHPND isolate S02 and the VP_AHPND isolate 5HP. The 2 major toxin bands (PirA$^{vp}$ at ~16 kDa and PirB$^{vp}$ at ~50 kDa) in lanes for partially purified 5HP proteins are absent from the proteins derived from S02.
(TIF)

**S3 Fig. The mRNA expression levels of the *Lv*APN2 gene in hemocyte of LvAPN1 knockdown shrimp.** Shrimp were injected with 0.85% NaCl, 20 μg/g shrimp of dsGFP, or 20 μg/g shrimp of dsAPN1 were determined by qRT-PCR. Relative expression of *Lv*APN2 gene is shown here in relative to that of EF-1α.
(TIF)

**S4 Fig. Recombinant protein purification analysis.** (A) SDS-PAGE analysis and (B) Western blot analysis with anti-His and anti-GST antibodies of recombinant truncated *Lv*APN1-His, rPirA$^{vp}$-His, rPirB$^{vp}$-GST and rGST protein overexpressed in *E. coli*. The His-tagged r*Lv*APN1 and rPirA$^{vp}$ were purified by Ni-NTA affinity chromatography. The deduced molecular weight for recombinant truncated *Lv*APN1-His and PirA$^{vp}$-His were 53 and 16 kDa, respectively. The rPirB$^{vp}$-GST fusion protein and rGST were purified by Sepharose 4B Glutathione beads. The estimated molecular weights for rPirB$^{vp}$-GST and rGST were approximately 70 and 23 kDa,

respectively.
(TIF)

## Acknowledgments

The authors acknowledge the Marine Shrimp Broodstock Research Center II (MSBRC-2), Charoen Pokphand Foods PCL for providing VP$_{AHPND}$. We thank Mr. Paul Barlow, National Cheng Kung University, for his helpful criticism of the manuscript.

## Author Contributions

**Conceptualization:** Waruntorn Luangtrakul, Pakpoom Boonchuen, Phattarunda Jaree, Han-Ching Wang, Kunlaya Somboonwiwat.

**Data curation:** Han-Ching Wang, Kunlaya Somboonwiwat.

**Formal analysis:** Waruntorn Luangtrakul.

**Investigation:** Waruntorn Luangtrakul.

**Methodology:** Waruntorn Luangtrakul, Pakpoom Boonchuen, Phattarunda Jaree, Ramya Kumar.

**Project administration:** Waruntorn Luangtrakul.

**Supervision:** Han-Ching Wang, Kunlaya Somboonwiwat.

**Validation:** Han-Ching Wang, Kunlaya Somboonwiwat.

**Visualization:** Kunlaya Somboonwiwat.

**Writing – original draft:** Waruntorn Luangtrakul.

**Writing – review & editing:** Han-Ching Wang, Kunlaya Somboonwiwat.

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
