## [Decision Letter · Decision Letter 0]

10 Jan 2021

Dear Dr. Somboonwiwat,

Thank you very much for submitting your manuscript "Cytotoxicity of Vibrio parahaemolyticus AHPND toxin on shrimp hemocytes, a newly identified target tissue, involves binding of toxin to aminopeptidase N1 receptor" for consideration at PLOS Pathogens. As with all papers reviewed by the journal, your manuscript was reviewed by members of the editorial board and by several independent reviewers. In light of the reviews (below this email), we would like to invite the resubmission of a significantly-revised version that takes into account the reviewers' comments. In particular, additional experimental data have been requested, as well as revisions within the text of manuscript to improve interpretability and clarity.

We cannot make any decision about publication until we have seen the revised manuscript and your response to the reviewers' comments. Your revised manuscript is also likely to be sent to reviewers for further evaluation.

Sincerely,

Steven R. Blanke

Associate Editor

PLOS Pathogens

Karla Satchell

Section Editor

PLOS Pathogens

Kasturi Haldar

Editor-in-Chief

PLOS Pathogens

orcid.org/0000-0001-5065-158X

Michael Malim

Editor-in-Chief

PLOS Pathogens

orcid.org/0000-0002-7699-2064

Reviewer's Responses to Questions

**Part I - Summary**

Reviewer #1: This manuscript provides compelling data that the aminopeptidase N1 (APN1) protein of shrimp hemocytes, stomach, and hepatopancreas serves as the receptor for the AHPND toxin from Vibrio parahaemolyticus (Vp). This was somewhat anticipated by prior work indicating the homology between the toxin and Cry toxin and the knowledge that APN1 serves as a receptor for Cry. This toxin is unique however in that it is a binary toxin and the data provided in this paper indicate that both components are part of the receptor binding interaction. This is also a significant interaction to have documented as the toxin is responsible for acute hepatopancreatic necrosis disease, which has significantly impacted shrimp production.

Reviewer #2: The title and abstract of the article are relevant and informative. The study method is valid and reliable with well-defined variables. The findings obtained were presented in an organized way and discussed from multiple angles and placed into the context without being over interpreted.

**Part II – Major Issues: Key Experiments Required for Acceptance**

Reviewer #1: While most of my comments are minor, it is important that the authors include data on how many replicates for each experiment. This was not consistently described.

While not essential for the conclusions of the paper, it would definitely have been nice to see a binding curve for truncated rLvAPN1 with the heterodimer.

Reviewer #2: Authors have not done paper review properly

1. The B Subunit of PirABvp Toxin Secreted from Vibrio parahaemolyticus Causing AHPND Is an Amino Sugar Specific Lectin

2. PirABVP Toxin Binds to Epithelial Cells of the Digestive Tract and Produce Pathognomonic AHPND Lesions in Germ-Free Brine Shrimp

Especially the later paper demonstrate that PirABVP toxin are not specific to hepatopancreas but it can target and binds to midgut and hindgut region of digestive tract.

So I think these paper can be used for writing more interesting discussion.

**Part III – Minor Issues: Editorial and Data Presentation Modifications**

Reviewer #1: Minor Points

1. Figures 1 and 2 are mis-ordered.

2. Move Fig S4 to Fig S3 to stay consider with order of presentation in manuscript..

3. Symbols in panel 3B almost completely illegible. Use color.

4. Fig. 3C. Use of two different types of arrow heads would be clearer than the # and * symbols in pointing the eye to the relevant features.

5. Lines 233-234: “Assuming a one-site binding model, the apparent dissociation constants (Kd) of truncated rLvAPN1 to rPirAvp and rPirBvp, as calculated from the saturation curves, were 3.157Å~10-6 M and 0. 499Å~10-6 M, respectively.”

I think stating the apparent Kd as 3.2 uM and 0.5 uM would be more appropriate as ELISA’s are not high precision measurements.

6. The cartoon in Figure 7 might lead readers to think that only the first toxin requires a receptor. Once one is bound, the remain toxins bind by oligomerization. Assuming this is not the intended implication, I recommend removing the receptor for the toxin once the toxin is drawn in its oligomerized form. Authors can note that it is not clear if the receptor remains bound.

7. Western blots using anti-His and anti-GST antibodies were used to confirm the expression of the respective proteins.

Authors should state the source of the antibodies. If commercial, include the company or catalog number.

Fig 2c. How many replicates were performed?

Fig. 1: How many replicates were performed?

Fig. 5: How many replicates were performed?

Reviewer #2: Line 78- they are homologous to Photorhabdus luminescens insect-related (Pir) toxins PirA/PirB not PirAvp/PirBvp.

Superscript Vp comes from Vibrio parahaemolyticus.

Line 354 Why for bacterial challenge and toxin challenge assay different AHPND and non-AHPND strains were used?

Line 349 If the strain is non-AHPND then how it will be VPAHPND strain, (MC)?

PLOS authors have the option to publish the peer review history of their article (what does this mean?). If published, this will include your full peer review and any attached files.

Reviewer #1: No

Reviewer #2: **Yes: **Vikash kumar
---

## [Editor Report · Decision Letter 1]

9 Mar 2021

Dear Dr. Somboonwiwat,

We are pleased to inform you that your manuscript 'Cytotoxicity of Vibrio parahaemolyticus AHPND toxin on shrimp hemocytes, a newly identified target tissue, involves binding of toxin to aminopeptidase N1 receptor' has been provisionally accepted for publication in PLOS Pathogens.

Best regards,

Steven R. Blanke

Associate Editor

PLOS Pathogens

Karla Satchell

Section Editor

PLOS Pathogens

Kasturi Haldar

Editor-in-Chief

PLOS Pathogens

orcid.org/0000-0001-5065-158X

Michael Malim

Editor-in-Chief

PLOS Pathogens

orcid.org/0000-0002-7699-2064
---

## [Editor Report · Acceptance letter]

23 Mar 2021

Dear Dr. Somboonwiwat,

We are delighted to inform you that your manuscript, "Cytotoxicity of Vibrio parahaemolyticus AHPND toxin on shrimp hemocytes, a newly identified target tissue, involves binding of toxin to aminopeptidase N1 receptor," has been formally accepted for publication in PLOS Pathogens.

Best regards,

Kasturi Haldar

Editor-in-Chief

PLOS Pathogens

orcid.org/0000-0001-5065-158X

Michael Malim

Editor-in-Chief

PLOS Pathogens

orcid.org/0000-0002-7699-2064